# Biopriming with Seaweed Extract and Microbial-Based Commercial Biostimulants Influences Seed Germination of Five *Abelmoschus esculentus* Genotypes

**DOI:** 10.3390/plants10071327

**Published:** 2021-06-29

**Authors:** Gugulethu Makhaye, Adeyemi O. Aremu, Abe Shegro Gerrano, Samson Tesfay, Christian P. Du Plooy, Stephen O. Amoo

**Affiliations:** 1Agricultural Research Council–Vegetables, Industrial and Medicinal Plants, Private Bag X293, Pretoria 0001, South Africa; makhayegugulethu@gmail.com (G.M.); AGerrano@arc.agric.za (A.S.G.); IduPlooy@arc.agric.za (C.P.D.P.); 2Discipline of Horticultural Science, School of Agricultural, Earth and Environmental Sciences, University of KwaZulu-Natal, Private Bag X01, Scottsville, Pietermaritzburg 3209, South Africa; Tesfay@ukzn.ac.za; 3Indigenous Knowledge Systems Centre, Faculty of Natural and Agricultural Sciences, North-West University, Private Bag X2046, Mmabatho 2790, South Africa; 4School of Life Sciences, University of KwaZulu-Natal, Private Bag X01, Scottsville, Pietermaritzburg 3209, South Africa; 5Crop Science Department, Faculty of Natural and Agricultural Sciences, School of Agricultural Sciences, North-West University, Private Bag X2046, Mmabatho 2790, South Africa

**Keywords:** biostimulants, food security, Kelpak^®^, imbibition, plant-growth-promoting rhizobacteria, seed priming

## Abstract

Seed germination is a crucial step in plant propagation, as it controls seedling production, stand establishment and ultimately crop yield. Approaches that can promote seed germination of valuable crops remain of great interest globally. The current study evaluated the effect of biostimulant (Kelpak^®^ and plant-growth-promoting rhizobacteria—PGPR) biopriming on the seed germination of five (VI037996, VI046567, VI055421, VI050956, and VI033796) *Abelmoschus esculentus* genotypes. The germination responses of the bio-primed seeds were measured using six parameters, including final germination percentage (FGP), mean germination time (MGT), germination index (GI), coefficient of velocity of germination (CVG), germination rate index (GRI), and time spread of germination (TSG). Biostimulant application significantly affected MGT (1.1–2.2 days), CVG (1.4–5.9), and TSG (1.2–3.0 days). Genotype also significantly influenced the TSG (1–3 days). Significant interaction effect of biostimulant treatment and genotype was evident on the FGP, GI, and GRI of the germinated seeds. The most noteworthy effect was demonstrated by Kelpak^®^ (1:100) applied to genotype VI037996, with significantly improved FGP (82%), GI (238), and GRI (77%/day) when compared to the control. Overall, the current findings suggest the potential stimulatory effect of biostimulants (especially Kelpak^®^) on the germination of *Abelmoschus esculentus* seeds. However, this influence was strongly dependent on the type of genotype.

## 1. Introduction

The increasing world population has resulted in an increasing demand for food, especially in developing countries. This increase has led to the promotion of multipurpose plant cultivation. However, the cultivation of these plants is largely dependent on seed germination, which in turn is influenced by several factors [1]. Seed germination is the initial step in a plant’s life and refers to the protrusion of the radicle from the seed coat [2]. Germination is an internal process that is facilitated by signalling pathways required to activate α-amylase and commence the breakdown of the starchy endosperm into simpler sugars to provide energy for the growing embryo [3].

Seeds of various plant species have different responses that are entirely based on the genetic make-up of that species [4,5]. Currently, the cultivation of many economically important plants is faced with poor seed germination rate, inadequate seedling emergence, and poor stand establishment, which negatively affects their yield [5]. As a result, it is pertinent to explore strategies to improve and synchronize seed germination as well as resultant field performance [6,7]. Recently, diverse novel technologies such as high-pressure processing, pulsed electric field, ultrasound, ozone processing, ultraviolet, magnetic field, microwave radiation, non-thermal plasma, electrolyzed oxidizing water, and plasma-activated water have been explored [7]. As a sustainable approach, researchers have investigated the effect of seed soaking and/or priming with various compounds that can promote germination [8,9,10,11]. Particularly, the application of biostimulants in the form of seed coatings that serve as a suitable delivery system has the potential to enhance germination rate and subsequent seedling establishment [12,13].

The stimulatory effect of biostimulants, such as seaweed extracts and plant-growth-promoting rhizobacteria (PGPR), on seed germination has been widely recognized [14,15]. Seaweed extracts are predominantly high in phytohormones, which tend to play a major role in seed germination [16,17]. Application of *Sargassum liebmannii* extracts significantly enhanced the germination percentage of *Trigonella foenum-graecum* seeds [18]. Furthermore, *Ascophylum nodosum* extract significantly enhanced the germination percentage and germination speed index of *Phaseolus vulgaris* [19]. Likewise, PGPR can synthesize phytohormones through their secondary metabolism [20]. *Bacillus subtilis* enhanced the germination percentage of *Sorghum bicolor* seeds, while *Panax schinseng* had no significant effect on the germination parameters of *Lactuca sativa* seeds [21,22].

*Abelmoschus esculentus* (L.) Moench (family: Malvaceae) is a rich source of important nutrients and phytochemicals, which makes it a valuable crop for combating nutrient deficiencies and promoting good health and well-being in humans [23,24,25]. However, low production and productivity of this species which has been attributed to several biotic and abiotic factors that may affect seed germination currently exists in sub-Saharan Africa [24]. Timely and uniform seed emergence has the potential to mitigate these biotic and abiotic challenges [6,26]. Sharma et al. [26] explored the effects of hydropriming, osmopriming, halopriming and solid matrix priming on *Abelmoschus esculentus* seeds as an approach for improving seed germination. Furthermore, organic amendments affect the germination and seedling vigour of *Abelmoschus esculentus* [27], while the potential of liquid-seaweed fertilizer, applied as a foliar spray, on the yield and nutritional quality of *Abelmoschus esculentus* was investigated by Zodape et al. [28]. Even though these aforementioned studies are laudable, the critical effects linked to different genotypes and biostimulants remain poorly understood. Thus, the current study was aimed at determining the effect of seed priming with Kelpak^®^ (a seaweed-based commercial biostimulant) and PGPR (a microbial-based biostimulant) on the various germination parameters of five *Abelmoschus esculentus* genotypes.

## 2. Results

### 2.1. Effect of Hydropriming Duration on Seed Germination

Final germination percentage, GI, and GRI increased with an increase in soaking duration (Table 1). The highest germination percentage was recorded in seeds soaked for 24 h. Mean germination time decreased with an increase in soaking duration. The lower the MGT, the faster the germination of a seed lot. Seeds soaked for 24 h had the lowest MGT, implying a faster germination in comparison to other soaking treatments. This soaking treatment also had the highest GI and GRI values. There was no significant difference in the TSG. On this basis, the 24 h soaking duration was used for the seed biopriming study.

### 2.2. Effect of Biopriming Treatments on Seed Germination of Abelmoschus esculentus Genotypes

Biostimulant biopriming treatments significantly affected all germination parameters, while the influence of genotype was evident only on FGP, GI, GRI, and TSG (Table 2). Biopriming with PGPR 1:5 (*v*/*v*) dilution significantly increased MGT, while other treatments had no significant influence when compared to the control (Figure 1). However, for both Kelpak^®^ and PGPR treatments, MGT decreased with an increase in dilution. Although the CVG increased with an increase in PGPR dilution, both biostimulants did not significantly increase CVG in comparison to the control (Figure 2). The lower the TSG value, the smaller the difference in germination speed of seeds within a seed lot [2]. Seeds primed with Kelpak^®^ 1:100 (*v*/*v*) had the lowest TSG value (Figure 3), implying a higher homogeneity in germination speed resulting from this treatment. The TSG value was also significantly influenced by genotype, with VI050956 having the highest homogeneity (Figure 4).

Table 3 presents the interaction effect of genotype and seed biopriming treatments on FGP, GI and GRI values. Generally, the FGP, GI, and GRI increased with an increase in biostimulant dilution, particularly for PGPR treatments. Seed priming with PGPR 1:5 (*v*/*v*) had a significantly negative effect on FGP, GI, and GRI. Kelpak^®^ 1:100 (*v*/*v*) significantly increased FGP, GI, and GRI in genotype VI037996, in comparison to the control.

## 3. Discussion

The benefits associated with the use of various priming agents, especially biostimulants, for enhancing seed germination cannot be over-emphasized in efforts geared towards improving food security and sustainability [6,29]. In particular, enhancing germination in a timely and uniform manner often exerts a beneficial effect on the seedling vigour and subsequent development of many plants with nutritional and medicinal values [8,9,10,11]. In the current study, a dose–response effect was evident in the evaluated germination parameters. The efficacy of seaweed extracts was affected by their concentration and in most cases, it enhanced plant growth attributes at low concentrations [30]. This study revealed the most noteworthy response in Kelpak^®^-treated (1:100, *v*/*v*) *Abelmoschus esculentus* seeds. This aforementioned treatment had a significantly positive effect on the FGP, GI, and GRI of genotype VI037996 when compared to the control. However, Kelpak^®^ (all dilutions) did not improve the MGT of *Abelmoschus esculentus* seeds. Kelpak^®^ is produced from *Ecklonia maxima*, which is one of the most commonly utilized seaweeds [17]. The positive effect of seaweed extract was demonstrated with the priming of *Phaseolus vulgaris* seeds using *Ascophyllum nodosum*, which resulted in a significantly increased germination speed index [19]. Furthermore, *Sargassum vulgare*, *Colpomenia sinuosa*, and *Padina pavonica* (all at 5% *v*/*v*) positively influenced the germination percentage of *Trigonella foenum-graecum* relative to the control [18]. The application of *Sargassum tenerrimum* (0.8% *v*/*v*) to *Solanum lycopersicum* seeds resulted in 100% germination, which was significantly higher than the control [29]. Brown seaweed (*Cystoseira barbata*) extracts significantly increased the germination percentage of *Solanum lycopersicum* and *Solanum melongena* seeds when compared to the control [31]. Seaweed extracts often stimulate and accelerate cell division, elongation, differentiation and protein synthesis [18], which likely explains the stimulatory effect of Kelpak^®^ on seed germination. In addition, the beneficial effects of seaweed extracts on seed germination have been attributed to the presence of phytohormones such as gibberellic acid and auxins [30,32,33]. Even though auxins do not directly affect seed germination, they facilitate gibberellic acid biosynthesis [5]. Gibberellic acid promotes germination, while auxins promote the biosynthesis of gibberellic acid, which therefore triggers the activities of ∝-amylase [34]. As postulated by Zulfiqar [6], improved germination of primed seeds may also be attributed to a solution–retention effect in the pre-germination phase that subsequently influences vital metabolic processes.

In this study, PGPR treatments did not significantly improve germination. Particularly, FGP, GI, and GRI in PGPR (1:5) treatment was significantly lower relative to the control. Several authors have documented varied germination responses in relation to the application of different rhizobacteria. The application of *Azospirillum brasilense* (Sp7, Sp7-S, and Sp245) to tomato seeds increased germination value, while in lettuce only the Sp7 strain promoted germination value relative to the control [34]. In *Allium cepa* seeds, *Bacillus* sp. and *Pseudomonas* sp. reduced germination (%), while *Azotobacter* sp. had no significant effect when compared to the control [35]. The authors elucidated that *Bacillus* sp. and *Pseudomonas* sp. produce hydrogen cyanide gas, which when available in larger quantities becomes toxic to the seeds and hence inhibits germination. Treatment of *Cuscuta campestris* seeds with *Bacillus* sp. had no significant effect on seed germination when measured against the control [36]. *Bacillus subtilis* promoted the germination (%) of sorghum var. CSH-14 and Proagro compared to the control [22]. *Brevibacillus brevis* significantly increased both the germination percentage and rate of *Gossypium hirsutum* seeds [37].

Plant genes play a major role in the germination process, particularly in the repair and synthesis of DNA during phase II of germination [5]. The genetic differences among cultivars cause different cultivar responses during this stage. While some cultivars complete phase II within a short period, some may require an extended period to complete it [38,39,40]. Similarly to the current study, varying germination responses among genotypes of different plants such as *Glycine max* [38], sugar beet [40] and *Jatropha curcas* [41] have been demonstrated. On the other hand, all 12 genotypes of *Triticum aestevium* had similar seed germination responses [39]. Contrary to the current study, the gene expression of these genotypes is almost similar (especially at the initial stages of growth), and all the phases of germination (imbibition, mobilization of food reserves, and radicle emergence) were in sync, with a high likelihood that seedling growth and stand establishment would be synchronised.

## 4. Materials and Methods

### 4.1. Source of Biostimulants and Seeds

Kelpak^®^ was obtained from Kelp Products (Pty) Ltd., Simon’s Town, South Africa. It is a commercial product that is made from *Ecklonia maxima* and known to contain diverse compounds such as polyamines, plant growth regulators, and phlorotannins [17]. Plant-growth-promoting rhizobacteria commercial solution (a mixture of organic acids, *Bacillus* sp., amino/fulvic acid, and soil bacteria) was purchased from Agriman (Pty) Ltd., Pretoria, South Africa. Five genotypes (VI037996, VI046567, VI055421, VI050956, and VI033796) of *Abelmoschus esculentus* seeds were obtained from the Agricultural Research Council–Vegetables, Industrial and Medicinal Plants (ARC-VIMP) gene bank, Pretoria, South Africa, which were previously imported from the World Vegetables Center, Taiwan, and conserved in the ARC-VIMP gene bank.

### 4.2. Hydropriming Duration Determination

The seeds of genotype VI037996 were surface-sterilized with a 1% sodium hypochlorite solution for 5 min and rinsed thoroughly with distilled water. Thereafter, they were soaked in distilled water for 0, 6, 12, and 24 h. For each soaking duration, 25 seeds were placed in 90 mm Petri dishes lined with two layers of filter paper (Whatman No. 1) and moistened with 10 mL distilled water. This was replicated three times. The Petri dishes were incubated in a growth chamber set at 25 °C with a 12/12 h light and dark regime. Germination was considered to be complete when the radicle had protruded at least 2 mm. Germination was monitored and recorded daily. Six germination parameters (final germination percentage (FGP), mean germination time (MGT), germination index (GI), coefficient of velocity of germination (CVG), germination rate index (GRI), and time spread of germination (TSG)) were calculated according to Kader [2] as follows:FGP = (Final number of seeds germinated in a seed lot/total number of seeds in a lot) × 100
MGT = Σ_*f.x*_/Σ_*f*_
CVG = *N*_1_ + *N*_2_ + · · · + *N_x_*/100 × *N*_1_*T*_1_ + · · · + *N_x_**T_x_*
GRI = G1/1 + G2/2 +· · ·+ G*_x/x_*
GI = (10 × n1) + (9 × n2) + · · · + (1 × n10)
TSG = the time in days between the first and last germination events occurring in a seed lot
where
*f* = seeds germinated on day *x*,*N* = number of seeds germinated each day,*T* = number of days from seeding corresponding to *N*,G1 = germination percentage on the first day after sowing,G2 = germination percentage on the second day after sowing,n1, n2 … n10 = number of germinated seeds on the first, second and subsequent days until the 10th day; 10, 9… and 1 are weights given to the number of germinated seeds on the first, second, and subsequent days, respectively.

### 4.3. Seed Priming with Biostimulants

Seeds of different genotypes were soaked for 24 h (based on the seed soaking experiment) in biostimulants at varying dilutions (Kelpak^®^ solution (1:20, 1:40 and 1:100, *v*/*v*) and PGPR (1:5, 1:10 and 1:15, *v*/*v*)), while distilled water was included as the control. For each treatment, 25 seeds were placed in 90 mm Petri dishes lined with two layers of Whatman No. 1 filter paper. The seeds were incubated at 25 °C with a 12/12 h light and dark regime for 14 days. The Petri dishes were laid out in a completely randomised design in triplicates. Seed germination was monitored and recorded daily.

### 4.4. Data Analysis

Bartlett’s Test as well as Levene’s Test of Equality of Variances was used to assess statistical assumption, and the Normtest procedure in Genstat software was used to assess the normality of the residuals for each variate in turn. The residuals were acceptably normal, but with some heterogeneous treatment variances. However, the analysis of variance (ANOVA) utilised the *t* and *F* statistics respectively, which are generally robust to violations of this assumption as long as group sizes are equal. Glass et al. [42] also indicated that the consequences on inference after ANOVA are not serious. Thus, data were subjected to a two-way ANOVA using Genstat 64-bit Release 18.2 (PC/Windows 8). Mean values were separated using Fishers’ protected t-test least significant difference (LSD) at the 5% level of significance [43].

## 5. Conclusions

In the absence of biostimulants, genotypes V1046567 and V1050956 demonstrated the most noteworthy germination responses. In relatively low-germinating genotypes, Kelpak^®^ application (1:100 dilution) significantly influenced the germination parameters of *Abelmoschus esculentus* genotype VI037996 when compared to the control (lacking biostimulants). It was evident that the efficacy of biostimulants depended on the genotype in addition to biostimulant concentrations. Diverse responses including stimulatory, inhibitory, and neutral effects were demonstrated for the different treatments. Overall, this study demonstrated the importance and contribution of biostimulant type and concentration on seed germination of *Abelmoschus esculentus*. Given that seed biopriming may potentially exert a carry-over effect on the seedling growth and yield as well as on biochemical parameters, further investigation focused on these aspects remains pertinent.

## Figures and Tables

**Figure 1 plants-10-01327-f001:**
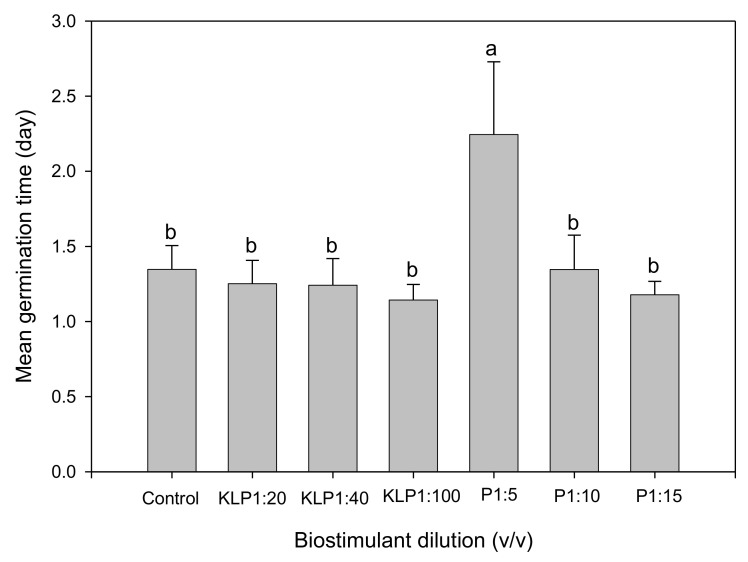
Effect of seed priming with biostimulants (KLP, Kelpak^®^; P, plant-growth-promoting rhizobacteria) on mean germination time of *Abelmoschus esculentus* seeds. Bars representing mean values and standard errors with different letters indicate statistically significant (*p* ≤ 0.05) differences based on Fisher’s protected least significant difference test (Genstat 64-bit Release 18.2).

**Figure 2 plants-10-01327-f002:**
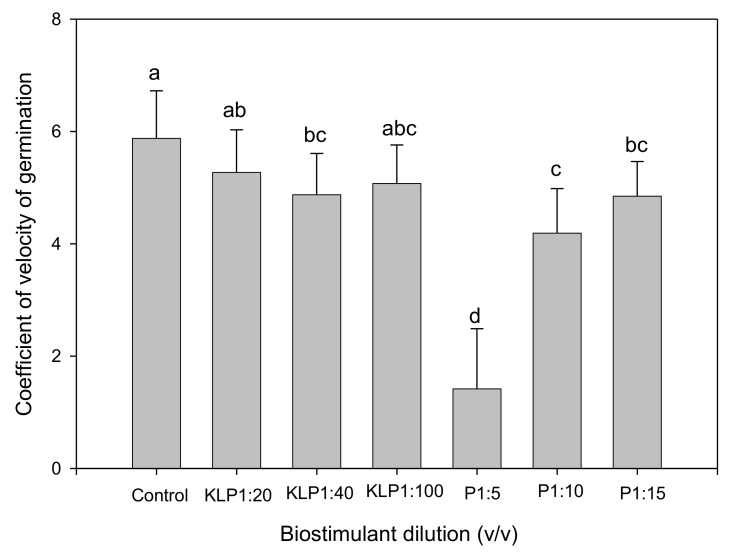
Effect of seed priming with biostimulants (KLP, Kelpak^®^; P, plant-growth-promoting rhizobacteria) on *Abelmoschus esculentus* seed coefficient of velocity of germination. Bars representing mean values and standard errors with different letters indicate statistically significant (*p* ≤ 0.05) differences based on Fisher’s protected least significant difference test (Genstat 64-bit Release 18.2).

**Figure 3 plants-10-01327-f003:**
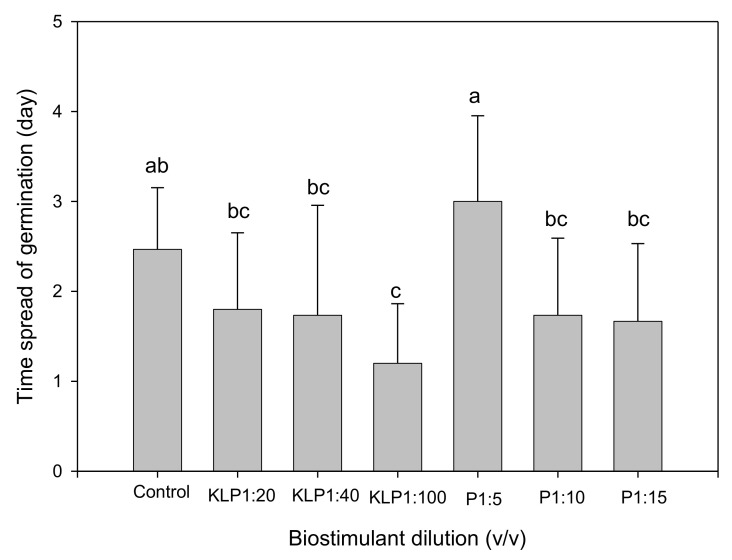
Effect of seed priming with biostimulants (KLP, Kelpak^®^; P, plant-growth-promoting rhizobacteria) on time spread of *Abelmoschus esculentus* seed germination. Bars representing mean values and standard errors with different letters indicate statistically significant (*p* ≤ 0.05) differences based on Fisher’s protected least significant difference test (Genstat 64-bit Release 18.2).

**Figure 4 plants-10-01327-f004:**
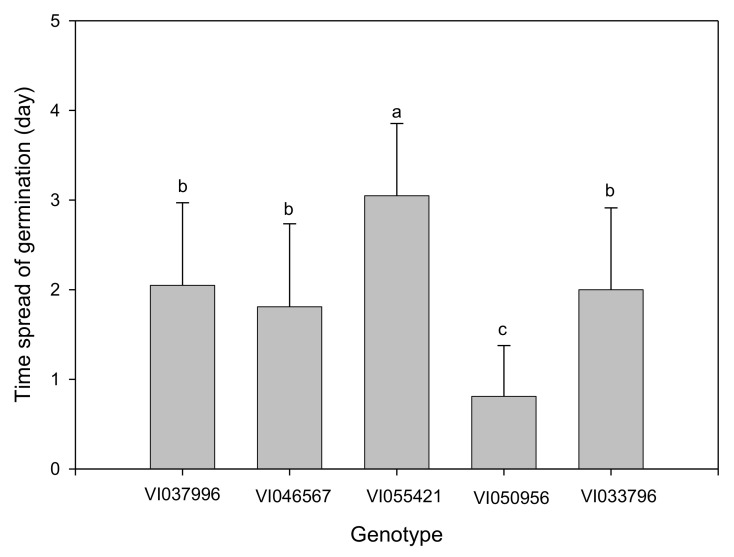
Effect of genotypes on time spread of *Abelmoschus esculentus* seed germination. Bars representing mean values and standard errors with different letters indicate statistically significant (*p* ≤ 0.05) differences based on Fisher’s protected least significant difference test (Genstat 64-bit Release 18.2).

**Table 1 plants-10-01327-t001:** Effect of hydropriming duration on *Abelmoschus esculentus* (genotype VI037996) seed germination incubated at 25 °C. FGP, final germination percentage; MGT, mean germination time; GI, germination index; CVG, coefficient of velocity of germination; GRI, germination rate index; TSG, time spread of germination. In each column, values (mean ± standard error) followed by different letters indicate statistically significant (*p* ≤ 0.05) differences. ns, not significant.

Hydropriming Duration (h)	FGP (%)	MGT (day)	GI	CVG	GRI (%/day)	TSG (day)
0	37.33 ± 7.06 ^b^	3.73 ± 0.20 ^a^	58.67 ± 11.84 ^c^	3.45 ± 1.09 ^b^	12.25 ± 2.70 ^c^	4.00 ± 0.58
6	62.67 ± 3.53 ^a^	3.61 ± 0.24 ^a^	99.67 ± 2.40 ^b^	9.05 ± 1.57 ^a^	19.25 ± 0.40 ^bc^	4.67 ± 0.88
12	60.00 ± 2.31 ^a^	2.66 ± 0.19 ^b^	110.00 ± 3.60 ^b^	6.04 ± 0.82 ^ab^	26.20 ± 1.89 ^b^	3.00 ± 0.58
24	72.00 ± 4.00 ^a^	2.52 ± 0.09 ^b^	134.67 ± 8.29 ^a^	8.20 ± 0.84 ^a^	38.09 ± 5.65 ^a^	5.00 ± 1.00
LSD (*p* ≤ 0.05)	14.91	0.6243	24.60	3.655	10.68	ns

**Table 2 plants-10-01327-t002:** Analysis of variance (ANOVA) for the effect of biostimulant and genotype on *Abelmoschus esculentus* seed germination. FGP, final germination percentage; MGT, mean germination time; GI, germination index; CVG, coefficient of velocity of germination; GRI, germination rate index; TSG, time spread of germination.

Source of Variation	df	MS					
		FGP	MGT	GI	CVG	GRI	TSG
Genotype (G)	4	358.75 ***	0.33 ns	6189.80 ***	0.87 ns	1019.47 ***	13.32 ***
Biostimulant treatment (B)	6	5963.53 ***	2.20 ***	75,466.80 ***	31.69 ***	6540.91 ***	5.32 *
G × B	24	333.31 ***	0.18 ns	4142.60 ***	2.76 ns	327.56 ***	1.63 ns
Residual	70	65.98	0.15	757.90	1.68	77.26	1.99
Total	104						

* *p* ≤ 0.05; *** *p* ≤ 0.001; ns, not significant; df, degrees of freedom; MS, mean squares.

**Table 3 plants-10-01327-t003:** Interaction effect of *Abelmoschus esculentus* genotypes and biostimulant (KLP, Kelpak^®^; PGPR, plant-growth-promoting rhizobacteria) treatments. FGP, final germination percentage; GI, germination index; GRI, germination rate index. In each column, mean values (±standard errors) followed by different letters indicate statistically significant (*p* ≤ 0.05) differences.

Genotype	Treatment	FGP (%)	GI	GRI (%/day)
VI037996	Control	69.33 ± 7.06 ^g–j^	235.00 ± 22.94 ^g–k^	57.93 ± 4.56 ^klm^
	KLP 1:20	74.67 ± 1.33 ^e–i^	250.30 ± 7.22 ^f–j^	60.95 ± 2.74 ^i–m^
	KLP 1:40	78.67 ± 3.53 ^c–h^	266.30 ± 7.22 ^d–h^	68.43 ± 2.35 ^g–l^
	KLP 1:100	82.67 ± 3.53 ^a–f^	286.30 ± 10.81 ^a–f^	76.67 ± 3.71 ^b–h^
	PGPR 1:5	64.00 ± 6.11 ^ij^	214.70 ± 16.48 ^ijk^	50.04 ± 2.07 ^m^
	PGPR 1:10	72.00 ± 4.62 ^f–j^	248.30 ± 14.77 ^f–j^	65.56 ± 2.26 ^h–l^
	PGPR 1:15	72.00 ± 2.31 ^f–j^	249.30 ± 7.79 ^f–j^	67.11 ± 1.46 ^h–l^
VI046567	Control	94.67 ± 5.33 ^a^	326.70 ± 19.34 ^a^	88.22 ± 6.88 ^abc^
	KLP 1:20	90.67 ± 3.53 ^abc^	315.70 ± 12.81 ^abc^	88.78 ± 3.91 ^ab^
	KLP 1:40	82.67 ± 1.33 ^a–f^	288.00 ± 4.16 ^a–f^	81.60 ± 1.22 ^a–g^
	KLP 1:100	88.00 ± 4.00 ^a–d^	304.70 ± 17.33 ^a–d^	83.56 ± 8.44 ^a–f^
	PGPR 1:5	17.33 ± 1.33 ^k^	54.70 ± 4.91 ^l^	12.00 ± 1.39 ^n^
	PGPR 1:10	78.67 ± 3.53 ^c–h^	270.70 ± 10.68 ^d–h^	71.11 ± 3.49 ^f–l^
	PGPR 1:15	92.00 ± 2.31 ^ab^	320.30 ± 8.11 ^ab^	89.11 ± 2.48 ^ab^
VI055421	Control	77.33 ± 4.81 ^d–h^	259.30 ± 215.68 ^e–i^	65.87 ± 4.73 ^h–l^
	KLP 1:20	78.67 ± 9.61 ^c–h^	268.30 ± 30.56 ^d–h^	68.31 ± 6.22 ^g–l^
	KLP 1:40	81.33 ± 1.33 ^b–g^	279.30 ± 6.01 ^b–g^	72.55 ± 5.12 ^d–j^
	KLP 1:100	74.67 ± 2.67 ^e–i^	258.30 ± 8.33 ^e–i^	71.15 ± 1.82 ^f–l^
	PGPR 1:5	25.33 ± 2.67 ^k^	78.70 ± 8.25 ^l^	13.53 ± 1.74 ^n^
	PGPR 1:10	61.33 ± 7.06 ^j^	199.70 ± 25.83 ^k^	46.89 ± 9.32 ^m^
	PGPR 1:15	78.67 ± 7.42 ^c–h^	270.00 ± 25.94 ^d–h^	71.82 ± 8.70 ^e–k^
VI050956	Control	94.67 ± 2.67 ^a^	330.00 ± 9.02 ^a^	92.44 ± 2.70 ^a^
	KLP 1:20	86.67 ± 3.53 ^a–e^	303.00 ± 12.29 ^a–e^	86.00 ± 3.46 ^a–e^
	KLP 1:40	88.00 ± 2.31 ^a–d^	307.30 ± 8.11 ^a–d^	86.67 ± 2.67 ^a–d^
	KLP 1:100	85.33 ± 1.33 ^a–e^	298.30 ± 4.84 ^a–e^	84.67 ± 1.77 ^a–f^
	PGPR 1:5	16.00 ± 6.11 ^k^	51.00 ± 21.28 ^l^	9.04 ± 4.57 ^n^
	PGPR 1:10	80.00 ± 4.62 ^b–g^	277.00 ± 16.19 ^b–g^	74.00 ± 5.03 ^c–i^
	PGPR 1:15	82.67 ± 7.06 ^a–f^	286.70 ± 26.77 ^a–f^	77.33 ± 11.62 ^b–h^
VI033796	Control	82.67 ± 1.33 ^a–f^	279.70 ± 5.92 ^b–g^	68.67 ± 4.70 ^g–l^
	KLP 1:20	80.00 ± 0.00 ^b–g^	275.00 ± 2.00 ^c–g^	70.89 ± 3.11 ^f–l^
	KLP 1:40	66.67 ± 2.67 ^hij^	226.70 ± 12.87 ^h–k^	59.05 ± 6.48 ^j–m^
	KLP 1:100	89.33 ± 2.67 ^a–d^	307.30 ± 8.19 ^a–d^	79.56 ± 5.01 ^a–h^
	PGPR 1:5	21.33 ± 5.33 ^k^	66.70 ± 12.68 ^l^	14.67 ± 2.15 ^n^
	PGPR 1:10	61.33 ± 11.39 ^j^	212.70 ± 39.40 ^jk^	57.33 ± 10.48 ^lm^
	PGPR 1:15	80.00 ± 2.31 ^b–g^	275.00 ± 7.23 ^c–g^	72.67 ± 2.52 ^d–j^
LSD (*p* ≤ 0.05)		13.23	44.83	14.31

## Data Availability

Data associated with the current study are presented in the manuscript.

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
