# Peer review of "Biopriming with Seaweed Extract and Microbial-Based Commercial Biostimulants Influences Seed Germination of Five Abelmoschus esculentus Genotypes"

_plants, 2021, doi:10.3390/plants10071327_

Round 1

Reviewer 1 Report

The manuscript is discussing the biopriming of 5 different genotypes of the Abelmoschus esculentus using biostimulants materials such as seaweed extract and microbial compounds. It is well-written and well-structured. The manuscript is prepared properly from the M&M to the results and fits the Plants journal scopes. My suggestion is to expand the introduction to at least a paragraph about other seed enhancing methods to introduce new methods to the audience for application to the seeds.  

LINE 53: May you please expand this section of the intro and use other modern technologies that are using biostimulants to enhance germination, stand establishment, and crop uniformity? I have some references from recent literature and encourage the authors to study for expanding of  introduction section: 

Amirkhani, M.; Mayton, H.S.; Netravali, A.N.; Taylor, A.G. A Seed Coating Delivery System for Bio-Based Biostimulants to Enhance Plant Growth. Sustainability 2019, 11, 5304. 

Ma, Y.; Látr, A.; Rocha, I.; Freitas, H.; Vosátka, M.; Oliveira, R.S. Delivery of inoculum of Rhizophagus irregularis via seed coating in combination with Pseudomonas libanensis for cowpea production. Agronomy 2019, 9, 33. 

LINE 186: May you please add a couple of sentences on the implication of your research for the future at the end of the discussion?

LINE 231. Please add info on the normality and variance homogeneity test of the data, any data transformation if you’ve done before subjecting the data to the ANOVA

LINE 259: Please follow the Plants journal’s format to prepare the references list.

Author Response

REVIEWER 1

The manuscript is discussing the biopriming of 5 different genotypes of the Abelmoschus esculentus using biostimulants materials such as seaweed extract and microbial compounds. It is well-written and well-structured. The manuscript is prepared properly from the M&M to the results and fits the Plants journal scopes. My suggestion is to expand the introduction to at least a paragraph about other seed enhancing methods to introduce new methods to the audience for application to the seeds.  

LINE 53: May you please expand this section of the intro and use other modern technologies that are using biostimulants to enhance germination, stand establishment, and crop uniformity? I have some references from recent literature and encourage the authors to study for expanding of  introduction section: 

Amirkhani, M.; Mayton, H.S.; Netravali, A.N.; Taylor, A.G. A Seed Coating Delivery System for Bio-Based Biostimulants to Enhance Plant Growth. Sustainability 2019, 11, 5304. 

Ma, Y.; Látr, A.; Rocha, I.; Freitas, H.; Vosátka, M.; Oliveira, R.S. Delivery of inoculum of Rhizophagus irregularis via seed coating in combination with Pseudomonas libanensis for cowpea production. Agronomy 2019, 9, 33. 

RESPONSE: As suggested, we have expanded the ‘Introduction’ to highlight the utilisation of novel technologies. In addition to the two suggested references, we have included the review paper by Rifna et al (2019)

Rifna, E.J., Ratish Ramanan, K., Mahendran, R. 2019. Emerging technology applications for improving seed germination. Trends in Food Science and Technology 86:95-108.

LINE 186: May you please add a couple of sentences on the implication of your research for the future at the end of the discussion?

RESPONSE: This concern has already been captured in the ‘Conclusion’ section of the manuscript. Inclusion in the 'discussion’ will result in unnecessary repetitions of the same information. In the revised manuscript, we have clearly indicated the genotype and treatment that can be considered as the most noteworthy based on our experiment.

LINE 231. Please add info on the normality and variance homogeneity test of the data, any data transformation if you’ve done before subjecting the data to the ANOVA

RESPONSE: We have updated the data analysis section to clearly indicate the approach and basis for this as supported by existing literature.

Snedecor, G., Cochran, W. 1980. Statistical methods. 7th edition. Iowa: The Iowa State University Press.

Glass, G.V., Peckham, P.D., Sanders, J.R. 1972. Consequences of failure to meet assumptions underlying the fixed effects analyses of variance and covariance. Review of Educational Research 42:237-288.

LINE 259: Please follow the Plants journal’s format to prepare the references list.

RESPONSE: All the references have been carefully checked to ensure adherence to the journal’s recommended format

Reviewer 2 Report

The paper presented for review discusses the effect of different factors (Kelpak and PGPR) on seed germination of 5 Abelmoschus esculentus genotypes. Research related to the effect of factors on seed germination is a very important and relevant issue from the point of view of application and improvement of annual production efficiency.

Below I send detailed comments on the manuscript.

Information related to the cultivation of Abelmoschus esculentus is lacking in the introduction. Is there a description in the scientific literature of research related to seed germination of this plant? It would be useful to expand this section of the manuscript to include such information. If there is little or no work on this plant it is worth noting in this part of manuscript. It is also advisable to include information on the novelty and innovativeness of the work submitted for review. The results obtained described in a concise manner.

Discussion of the paper should be rewritten.

Discussion of the work refers to different plants that were stimulated often with other biostimulants or PGPR. Authors should focus on comparing the obtained results with scientific reports related to the use of Kelpak or components of this product. A discussion of the results of the study in relation to other scientific studies related to the cultivation of Abelmoschus esculentus is worthwhile.

The methodology of the study - the section on hydropriming duration determination - should be completed with information on which genotype of plants was used or if all 5 then this should be included.

Results of the study. The part about Efect of Hyropriminig Duration on seed germination, here we still don't know the seeds which phenotype are concerned. From table 2 we learn that genotype was taken into account, but what/which?

Only in Fig.4 we find information about the differences obtained for the genotypes.

Is it known which strains of soil bacteria were present in the preparation obtained from Agriman (Pty) Ltd?

I am also wondering about the title of the paper? It is not quite adequate to the research that has been carried out, because commercial preparations were analysed, the composition of which is not only sedweed extract and microbial biopriming.

Author Response

REVIEWER 2

The paper presented for review discusses the effect of different factors (Kelpak and PGPR) on seed germination of 5 Abelmoschus esculentus genotypes. Research related to the effect of factors on seed germination is a very important and relevant issue from the point of view of application and improvement of annual production efficiency.

Below I send detailed comments on the manuscript.

Information related to the cultivation of Abelmoschus esculentus is lacking in the introduction. Is there a description in the scientific literature of research related to seed germination of this plant? It would be useful to expand this section of the manuscript to include such information. If there is little or no work on this plant it is worth noting in this part of manuscript. It is also advisable to include information on the novelty and innovativeness of the work submitted for review. The results obtained described in a concise manner.

RESPONSE: We have briefly highlighted existing information regarding the germination of okra. This has been supported with the new references listed below. In addition, we have highlighted the novelty of our work in the revised manuscript (see sentence highlighted in red font in the introduction)

Sarma, B., Gogoi, N. 2015. Germination and seedling growth of okra (Abelmoschus esculentus L.) as influenced by organic amendments. Cogent Food and Agriculture 1:1030906.

Sharma, A.D., Rathore, S.V.S., Srinivasan, K., Tyagi, R.K. 2014. Comparison of various seed priming methods for seed germination, seedling vigour and fruit yield in okra (Abelmoschus esculentus L. Moench). Scientia Horticulturae 165:75-81.

Benchasri, S. 2012. Okra (Abelmoschus esculentus (L.) Moench) as a valuable vegetable of the world. Ratarstvo and Povrtarstvo 49:105-112.

Zodape, S., Kawarkhe, V., Patolia, J., Warade, A. 2008. Effect of liquid seaweed fertilizer on yield and quality of okra (Abelmoschus esculentus L.). Journal of Scientific and Industrial Research 67:1115-1117.

Discussion of the paper should be rewritten.

Discussion of the work refers to different plants that were stimulated often with other biostimulants or PGPR. Authors should focus on comparing the obtained results with scientific reports related to the use of Kelpak or components of this product. A discussion of the results of the study in relation to other scientific studies related to the cultivation of Abelmoschus esculentus is worthwhile.

RESPONSE: In the context of our experiment design, the discussion aligns with our findings in relation to existing studies. The discussion is in relation to the applied biostimulants and the plant studied. This is evident in the composition of our references in the discussion section. The discussion has sufficient depth in our opinion and we decided to keep this focus in line with our defined objective. 

The methodology of the study - the section on hydropriming duration determination - should be completed with information on which genotype of plants was used or if all 5 then this should be included.

RESPONSE: We have indicated in section 4.2 that the seeds of genotype VI037996 was used for this preliminary check.

Results of the study. The part about Efect of Hyropriminig Duration on seed germination, here we still don't know the seeds which phenotype are concerned. From table 2 we learn that genotype was taken into account, but what/which?

RESPONSE: The effect of hydro-priming serves as a quick preliminary check and we only tested 1 genotype (VI037996). This has been highlighted in the revised manuscript.

Only in Fig.4 we find information about the differences obtained for the genotypes.

RESPONSE: Yes, this is a scientifically correct approach as the result from the current research must be read in conjunction with statistic data in Table 2. The approach is to first consider significant interaction effect for the six parameters studied (in this case, FGP, GI and GRI were significant and presented in Table 3). In the absence of significant interaction effect, we presented the effect either due to the biostimulant (in this case, MGT, CVG and TSG were significant and presented as Figures 1, 2 and 3, respectively) or genotype (in this case, TSG was significant and presented as Fig. 4) effects.

Is it known which strains of soil bacteria were present in the preparation obtained from Agriman (Pty) Ltd?

RESPONSE: We have highlighted the details (i.e. Bacillus sp with other ingredients) as contained in the product we tested. Please see section 4.1 (line 3)

I am also wondering about the title of the paper? It is not quite adequate to the research that has been carried out, because commercial preparations were analysed, the composition of which is not only sedweed extract and microbial biopriming.

RESPONSE: The basis for our research involve the evaluation of existing biostimulants available to diverse group of farmers in South Africa. We decided to explore the potential of these products as a priming agent for seed germination of okra. In our opinion, testing of these commercial products (a form of re-purposing) using an important crop (okra) is valid. It is also advantageous that the products which are currently available on commercial scale may offer additional benefits to farmers. The composition of the products are mainly seaweed (Kelpak) and PGRP commercial solution (Bacillus sp). The diverse composition of biostimulants is well-known among scientists and other stakeholders (Yakhin et al 2017). As a response to the concern about the title, we have amended the title to “Biopriming with Seaweed Extract and Microbial-based Commercial Biostimulants Influence Seed Germination of Five Abelmoschus esculentus Genotypes”

Stirk, W.A., Rengasamy, K.R., Kulkarni, M.G., van Staden, J. 2020. Plant Biostimulants from Seaweed: An Overview. Pages 33-55 in Geelen, D. and Xu, L., editors. The Chemical Biology of Plant Biostimulants.

Yakhin, O.I., Lubyanov, A.A., Yakhin, I.A., Brown, P.H. 2017. Biostimulants in plant science: A global perspective. Frontiers in Plant Science 7:doi:10.3389/fpls.2016.02049.

Reviewer 3 Report

No comments. It should be publish as in present form. 

Author Response

Dear Reviewer,

Thank you for the positive response.

Reviewer 4 Report

I read with interest of manuscript "Seaweed Extract and Microbial Biopriming Influence Seed Germination of Five Abelmoschus esculentus Genotypes".
The results proposed are in part new. The mauscript is interesting because of its practical significance.

The MS, however, presents several lacks.
The composition of Kelpak and PGPR should be provided. If it has not been tested, I suggest you look for it in the literature.
The standard deviation should be provided in tables and figures.

The conclusions should specify which genotype and which treatment is best for use on farms. 

Author Response

REVIEWER 4

I read with interest of manuscript "Seaweed Extract and Microbial Biopriming Influence Seed Germination of Five Abelmoschus esculentus Genotypes".

The results proposed are in part new. The mauscript is interesting because of its practical significance.

The MS, however, presents several lacks.

The composition of Kelpak and PGPR should be provided. If it has not been tested, I suggest you look for it in the literature.

RESPONSE: In the current study, we did not analyse both products (outside the scope of our research) and have rather highlighted their composition based on existing literature and manufacturers’ descriptions (see font highlighted in red in section 4.1)

Kelpak is produced from the seaweed Ecklonia maxima and known to contain diverse components such as plant growth regulators and polyamines. 

Plant growth promoting rhizobacteria commercial solution is a mixture of organic acids, Bacillus sp., amino/fulvic acid, and soil bacteria.

Stirk, W.A., Rengasamy, K.R., Kulkarni, M.G., van Staden, J. 2020. Plant Biostimulants from Seaweed: An Overview. Pages 33-55 in Geelen, D. and Xu, L., editors. The Chemical Biology of Plant Biostimulants.

Yakhin, O.I., Lubyanov, A.A., Yakhin, I.A., Brown, P.H. 2017. Biostimulants in plant science: A global perspective. Frontiers in Plant Science 7:doi:10.3389/fpls.2016.02049.

The standard deviation should be provided in tables and figures.

RESPONSE: We have included the standard errors in the Tables and Figures in the revised manuscript.

The conclusions should specify which genotype and which treatment is best for use on farms.

RESPONSE: We have highlighted the genotype and treatment with the most noteworthy response based on our experiment. Please see the revised section 5 where we have highlighted the new additions in red font.

Round 2

Reviewer 2 Report

Thank you so much for your careful analysis of my review. It seems to me that the changes made to the manuscript, as well as the authors' detailed explanation of my questions/doubts, are sufficient to accept and publish the manuscript.